# Active Materials for 3D Printing in Small Animals: Current Modalities and Future Directions for Orthopedic Applications

**DOI:** 10.3390/ijms23031045

**Published:** 2022-01-18

**Authors:** Parastoo Memarian, Elham Pishavar, Federica Zanotti, Martina Trentini, Francesca Camponogara, Elisa Soliani, Paolo Gargiulo, Maurizio Isola, Barbara Zavan

**Affiliations:** 1Department of Animal Medicine, Productions and Health, University of Padova, 35020 Padova, Italy; parastoo.memarian@phd.unipd.it (P.M.); maurizio.isola@unipd.it (M.I.); 2Department of Translational Medicine, University of Ferrara, 44121 Ferrara, Italy; elham.pishavar@unife.it (E.P.); Federica.zanotti@unife.it (F.Z.); martina.trentini@unife.it (M.T.); Francesca.camponogara@unife.it (F.C.); 3Engineering Department, King’s College, London WC2R 2LS, UK; elisasoliani@college.uk; 4Institute for Biomedical and Neural Engineering, Reykjavík University, 101 Reykjavík, Iceland; paologar@landspitali.is; 5Department of Science, Landspítali, 101 Reykjavík, Iceland

**Keywords:** 3D printing, receiver-specific, veterinary, orthopedics, materials

## Abstract

The successful clinical application of bone tissue engineering requires customized implants based on the receiver’s bone anatomy and defect characteristics. Three-dimensional (3D) printing in small animal orthopedics has recently emerged as a valuable approach in fabricating individualized implants for receiver-specific needs. In veterinary medicine, because of the wide range of dimensions and anatomical variances, receiver-specific diagnosis and therapy are even more critical. The ability to generate 3D anatomical models and customize orthopedic instruments, implants, and scaffolds are advantages of 3D printing in small animal orthopedics. Furthermore, this technology provides veterinary medicine with a powerful tool that improves performance, precision, and cost-effectiveness. Nonetheless, the individualized 3D-printed implants have benefited several complex orthopedic procedures in small animals, including joint replacement surgeries, critical size bone defects, tibial tuberosity advancement, patellar groove replacement, limb-sparing surgeries, and other complex orthopedic procedures. The main purpose of this review is to discuss the application of 3D printing in small animal orthopedics based on already published papers as well as the techniques and materials used to fabricate 3D-printed objects. Finally, the advantages, current limitations, and future directions of 3D printing in small animal orthopedics have been addressed.

## 1. Introduction

There is a growing demand in veterinary practice to improve the animal’s quality of life. Corrective osteotomies, limb sparing, and joint-replacement surgeries are amongst the veterinary orthopedic procedures that tend to boost the animal’s limb function and, as a result, the animal’s quality of life [1,2,3,4]. In spite of this, many orthopedic conditions in animals are still treated with limb amputations and salvage procedures, resulting in reduced or impaired limb function and decreased quality of life [5]. Complex orthopedic disorders such as extensive bone loss, fracture nonunion or malunion, tumors, bone deformities, and large-scale traumatic injuries remain clinical challenges in veterinary practice as conventional surgical techniques usually fail to address these conditions [6,7,8]. The idea of bone tissue engineering (BTE) has been developed to resolve the limitations of conventional approaches in addressing complicated orthopedic circumstances [9,10,11]. An emerging approach in BTE is the construction of three-dimensional (3D) synthetic structures produced for a receiver-specific need with the possibility of cell and protein integration [10].

The introduction of modern developments in veterinary applications has led to substantial growth in general animal care and individualized veterinary medicine. Recently, 3D printing has emerged as a valuable approach in the manufacturing of individualized implants for receiver-specific needs in human and veterinary orthopedics [12,13]. The term “receiver-specific” describes the unique bone geometry or bone density of the receiver that is assessed utilizing the receiver’s medical images [14]. The necessity for receiver-specific therapy and custom-made implants is even more remarkable in veterinary practice because there are more size and geometric variations between and within different breeds and species of veterinary receivers compared with human receivers [8,12].

The features of using 3D-printed orthopedic implants have been described and used mainly in human medicine, and only a few publications include an evaluation of its potential application in veterinary orthopedics [12]. Three-dimensional printing technology enables the design of receiver-specific implants and their production from biocompatible or bioinert material with a relatively short lead-time from design to surgery [14]. This article has reviewed the AM approaches and the specifics of manufacturing 3D-printed materials in veterinary orthopedic surgery. The benefits and limitations of 3D printing technologies and their employment in veterinary orthopedic applications were discussed. The feature addressed in this review is mainly focused on small animals, particularly for use in dogs, but it is also applicable to other animals in common veterinary practice.

## 2. 3D Printing Process

Three-dimensional printing refers to the process of creating physical objects from digital models [14,15]. A virtual 3D design file is created by computer-aided design (CAD) software using a 3D modeling program, a 3D scanner, or medical scanning techniques. The CAD data are then converted into multiple 2D cross-section layers. Following the predefined 2D pattern, a 3D printer manufactures a 3D structure without any need for an intermediate molding step [14,16]. The advantages of 3D printing include design liberty, automation, manufacturing velocity, accuracy, customization, and minimal waste generation [11,14]. The technique of producing a 3D-printed object consists of three critical steps, including data acquisition, image processing, and 3D printing of object [14,16] (Figure 1).

### 2.1. Image Data Acquisition

The bony structure must be accurately captured to represent the individual receiver’s anatomy. The most common imaging modalities for obtaining medical information are computed tomography (CT) and magnetic resonance imaging (MRI), which provide fast and precise 3D image data with high resolution [11,16,17]. In orthopedic applications, image capturing is based primarily on CT images due to their high contrast and the ability to present accurate bone dimensions reliably. However, the broad range of animals’ sizes and the need to anesthetize the receiver during each imaging session are two main limitations of CT imaging in animals [8,16]. Most of the current CT machines export the acquired medical images as cross-sectional images in DICOM (digital imaging and communications in medicine), a standard data format to store, exchange, and transmit medical images. Therefore, in orthopedics, DICOM images are the critical connection between 3D printing technologies and receiver-specific medical imaging records. The DICOM Standards Committee is split into many subcommittees (Figure 2).

### 2.2. Image Processing

The image processing process require the use of a software able to produce special images, called DICOM, for the rendering and creation of the 3D mesh [16]. To obtain this, the acquired data are transferred as DICOM-compliant files in commercial or open-source 3D software programs for 3D object fabrication. With these programs, thin slices of axial images are used through the multiplanar reconstruction (MPR) technique to generate nonaxial 2D images [14].

In order to improve the clinical interpretation and visualization of complex 3D structures, additional reformatted coronal or sagittal images can then be visualized. This technique is beneficial for analyzing joint alignments and skeletal structures in fractures or limb deformities as specific details on axial parts may not be readily apparent [14]. To create 3D mesh models of the data set, other 3D simulation techniques such as volume rendering are used. Three-dimensional renders of DICOM images are made, which can be used for medical diagnostics or as a source for creating CAD files. The main benefits of 3D renderings are their low cost (once the software is purchased) and immediate availability. Renderings are also used in orthopedic receivers to improve the diagnosis and communication with receivers or pet owners. Scanning parameters, comprising the radiation intensity, slice thickness, and CT reconstruction algorithms, also affect the accuracy of 3D renderings [8,14]. Mimics Innovation Suite (Materialise, Leuven, Belgium), Geomagic Studio (Raindrop Geomagic. Research Triangle Park, NC, USA), and ScanIP (Synopsys, CA, USA) are some of the commercial packages that have been used for medical applications. Meshlab (Meshlab, ISTI—CNR research center, Pisa, Italy), InVesalius (Research Centre, of the Ministry of Science and Technology in Campinas, Campinas, Brazil), and Slicer (3-D Slicer, http://www.slicer.org, accessed on 10 January 2022) provide open-source applications [13,16].

#### 2.2.1. Image Segmentation

The first step toward 3D printing is to segment the DICOM images and create an STL model. After importing DICOM files, it is usually necessary to isolate and extract the structure of interest (bone) within the image data, typically calling for segmentation as a step to support 3D printing. This step uses specific density (thresholding) and topography of the regions of interest to isolate them and remove any unwanted or nonanatomic data, such as soft tissues or dense feces in the colon. Thresholding the magnitude of the voxel intensity is a common technique for segmenting regions, such as bones, with consistent differences in intensity from their surrounding tissues. Several thresholds can be set to display only pixels with gray values in the target range. Finally, 3D models can be created from the segmented areas of interest [14,16,18]. Primary processing can begin after exporting the data to a 3D CAD compatible file format, such as the intermediate data STL file format. Though DICOM images are not instantly converted to STL files, segmentation filters are. The quality of the STL data is proportional to the quality of the 3D model, and inaccurate STL data will result in the fabrication of a low-quality 3D model. After primary and secondary processing, including noise reduction and hole adjustment, the STL data can be 3D-printed. To create a 3D mesh model, a continuous mesh free of holes or faults in the rendered model is required. Using CAD tools, structures’ profiles are separated into various polygons, typically triangles, which may vary from 30,000 to millions, depending on the scale and the model complexity. The number of polygons is directly proportional to the resolution; raising the number of polygons results in a sharper and more precise surface, but it significantly increases the data size and causes further delays in processing [16,18]. When reverse-engineering scanners are used to obtain a data point package, similar software creates STL archives. For segmenting DICOM images to STL information, several medical programs including Mimics and 3D Slicer can be used [16].

#### 2.2.2. Manipulation and Analysis

DICOM data can be imported directly into the program for image manipulation, where it can be converted to a standard 3D file format, STL. Further editing of STL files, such as triangular mesh optimization or object geometry adjustment, is an optional step and can be performed before the CAD data are sent to a 3D printing machine for object fabrication. If any object, regardless of how it was built, needs to be modified in shape or form, programs such as Autodesk (Autodesk Inc., San Rafael, CA, USA) and freeware such as BRL-CAD (https://brlcad.org/, accessed on 10 January 2022) or Openscad (https://openscad.org/, accessed on 10 January 2022) are available [8,14,16].

### 2.3. Object Fabrication

The essential step in 3D model fabrication is creating the STL data. The final action is to import the STL files into proprietary applications connected with the printer or commercial applications, such as Fusion 360 with Netfabb^®^ (https://www.autodesk.com/products/netfabb/overview, accessed on 10 January 2022) and KISSlicer (https://www.kisslicer.com/, accessed on 10 January 2022) or multi-printer-compatible open-source applications (e.g., ReplicatorG, http://replicat.org/, accessed on 10 January 2022). It is vital to ensure that the program chosen for the final stage is compatible with the printer to be used [16]. The 3D printing of the physical 3D model necessitates using “G-code” generation software to generate G-code as 3D printable data. The STL files representing the 3D model will be processed and “sliced” into cross-sectional layers by CAD software. By inserting sequential layers of material to recreate the virtual cross-sections, a 3D printing system then manufactures the 3D physical model. Nevertheless, the accuracy of the final 3D model is influenced by each phase of the process, including DICOM image segmentation, STL data processing, G-code data generation, and 3D printer output [14,18].

## 3. Materials for 3D Printing

Numerous studies are now being conducted on developing new biomaterials for 3D printing [11]. Biomaterials are natural or manufactured substances that interact with biological tissues to help restoring and replacing tissues or organs. The choice of biomaterial type used in 3D printing is determined by the intended application of the final product [19,20]. For instance, a biomaterial for 3D printing in orthopedics should be easily printable and have excellent biocompatibility, controlled long-lasting biodegradation, acceptable mechanical properties, and a well-designed architecture [6,19,20]. For surgical application, the material should be sterilizable [14]. Even though 3D printing has been successfully used in various medical applications, the number of 3D-printable materials currently available is extremely limited. Titanium (Ti6Al4V) alloy, cobalt–chrome (CoCr) alloy, stainless-steel (SS) alloy, ceramics, polyetheretherketone (PEEK), and ultra-high-molecular-weight polyethylene (UHMWPE) are the most utilized biocompatible and implantable materials for 3D printing orthopedic applications [11]. Based on the chemical composition, 3D-printed biomaterials are roughly classified into four groups [20]. Table 1 summarizes the 3D-printable biomaterials and their applications in veterinary orthopedics.

### 3.1. Metals

The metals show the best properties to be selected as materials for a bone implant thanks to their low stiffness shape memory elastic behavior. Moreover, their physical properties, such as alloy fracture toughness (comprise from 55 to 60 MPa m1/2 in pure Ti), elastic modulus (comprise from 100 to 105 GPa), and compressive strength (from 130 to 170 MPa), also help their application for tissue engineering. Another important parameter is the porosity, which reduces the elastic modulus of implants but also influences the bone regeneration inside the scaffolds.

The best shape memory properties are of two alloys: copper-aluminum-nickel and nickel-titanium (Ni-Ti). The biological success of metal implants, namely to induce bone osteointegration, is related to its ability to promote cell attachment, proliferation, commitment, and mineralization of the extracellular matrix. This is affected by the metal surface properties, where topography can be modified through laser treatment at micro- and nanometer scales. In addition, coating with ceramics (silica, boron oxide, or other alkali metal oxides) increases the biological properties of the surface [11,14]. The material of cobalt alloys is tough, and the surface chromium oxide layer provides corrosion resistance. Nevertheless, because of its hardness, creating complex shapes and its machining are challenging [21,22]. Commercial titanium is highly biocompatible and corrosion resistant; however, it lacks sufficient mechanical strength. Ti6Al4V, the commonly used alloy in the manufacturing of biomedical components, is more potent than pure titanium and has superior fatigue resistance [23,24]. Tantalum is a metal that can be fabricated with a porosity and elastic modulus similar to cancellous bone. It is a unique, corrosion-resistant metal with excellent bone ingrowth characteristics that is increasingly being employed in surgical implants. However, tantalum machining is relatively expensive and challenging. Tantalum endo-prosthesis is a viable implant for limb-sparing surgery in dogs due to its better biocompatibility and mechanical characteristics over standard SS limb-sparing plates [12,25]. However, the trabecular tantalum endoprosthesis has lately been questioned for its possible role in increasing the risk of infection [26,27]. Porous metallic biomaterials are frequently used in several therapeutic applications, including joint replacement surgery. They have a large surface area and an open cell microarchitecture, which promotes bone ingrowth, biological fixation, and nutrient flow.

### 3.2. Ceramics and Glasses

Recently, different bioceramics have been employed for the construction of 3D-printed scaffolds or implants due to their promising osteoinductive and osteoconductive properties, high stiffness (393 GPa), and similarity to the mineral phase of bone [6,28]. Calcium phosphate ceramics (CaP ceramics) are synthetic materials made of calcium hydroxyapatites (HA) and thus have a composition comparable to the original bone matrix. Tricalcium phosphate (TCP), HA, and biphasic calcium phosphate (BCP) are the most often used CaP ceramics in bone reconstruction [6,29]. Bioactive glasses, often known as bioglasses, are ceramics made from synthetic silicates; they are rapidly resorbed in the first two weeks after implantation, allowing for rapid new bone formation and implant ingrowth. CaP ceramics and bioactive glasses have been used to make 3D-printed scaffolds; however, their poor mechanical properties, such as low fracture-toughness and tensile strength, limit their application in load-bearing conditions. This disadvantage can be overcome prior to printing by mixing bioceramics with polymers such as cellulose, poly (D, L-lactic acid-co-glycolic acid), or polycaprolactone (PCL), or by combining with specialized reinforcing materials such as carbon nanotubes, graphene, polyethylene, Al_2_O_3_, and TiO_2_ to create ceramic composites with higher mechanical strengths [6,7,28]. Graphene and its derivates showed to be promising materials in the field of material science due to their good biocompatibility and osteogenesis properties [6,30,31,32,33]. Our research group studied the biological and osteogenic properties of graphene-based scaffolds loaded with canine adipose-derived mesenchymal stem cells (cAD-MSCs) in vitro (Figure 3e–g). The results showed the excellent biocompatibility of graphene-based scaffold and the capability of carbon to improve the cell adhesion, growth, and osteogenic differentiation of cAD-MSCs. We proposed graphene-based scaffolds as innovative materials for BTE in veterinary practice [6]. Ceramics have also been used to reduce the micromotion between bone and implants during physiological load-bearing and to promote osteointegration between bone tissues and implants. Ceramic femoral heads have not been used in veterinary hip implants; however, femoral head coating with diamond-like carbon was used in the current generation of the Zurich cementless hip [21].

### 3.3. Polymers

Polymers are one of the most common materials used in 3D-printed bone replacements because of their potential use as filaments for fused deposition modeling (FDM), solutions for stereolithography apparatus (SLA), powder beads for selective laser sintering (SLS), and gels for direct ink writing (DIW). Poly-D, L-lactide (PDL), polylactic acid (PLA), polyglycolic acid (PGA) or their copolymers, and polylactic-co-glycolic acid (PLGA) are amongst the biodegradable polymers that have been utilized in 3D printing [34]. Polycaprolactone (PCL) is another biodegradable polymer with FDA approval widely used in 3D printing. Its excellent biocompatibility, slow degradation, and suitable mechanical properties make PCL one of the preferred polymers for manufacturing 3D-printed bone scaffolds [7,35]. High-density polyethylene (HDPE) is a thermoplastic polymer widely used in biomedical engineering. It is an economical material with good mechanical and thermal properties. Historically, commercially available veterinary total hip prostheses have bearing surfaces of polyethylene on the acetabular side [35]. UHMWPE was the first and is still the most commonly used polyethylene in veterinary joint replacements [21]. Developments in joint replacement surgeries in human total hip replacement (THR) contain considerable changes in material choices, particularly the move from HDPE to UHMWPE [35]. In addition, thermoplastic polymers such as PLA and acrylonitrile butadiene styrene (ABS) can also be used to fabricate the 3D models of limbs as well as receiver-specific surgical guides, instruments, and prosthesis [8,11]. Furthermore, polyamides have good stability, rigidity, flexibility, and shock-resistance properties. Recent studies have used polyamides in conjugation with HA to produce porous scaffolds with high load-bearing capabilities for bone regeneration [36].

Currently, 3D-printed hydrogel scaffolds have shown great promise creating customized scaffolds for BTE due to their excellent elasticity, tunable mechanics, biocompatibility, and biodegradability. The most common 3D printing methods are inkjet, laser-assisted bioprinting, and extrusion for hydrogel and bioinks. Cell viability and the stability of a vascularization network within the scaffolds are considered the major challenge that 3D printing methods can influence on cell viability. Previous studies have indicated that extrusion-based bioprinting leads to 98% of cell viability [37]. Dynamic structure hydrogels might enhance healable feature polymers at the molecular level so that they can reduce cell damage during extrusion printing. The self-repairing performance of these structures is characterized by the healing efficiency and the number of successful healing cycles that follow each other [38]. DIW is considered the most common method for the manufacture of self-healing smart structures [39].

### 3.4. Composites

Composites are manufactured materials made up of two or more elements with distinct physical characteristics that may be mixed in a synergistic manner. Composites can be highly biocompatible while keeping appropriate mechanical qualities due to the immense diversity of material structures that may be created [20,28]. Some composites have also been studied for 3D printing application in bone, including PCL/TCP, PLGA/TCP/HA, and PCL/PLGA/TCP. For instance, PCL has been combined with β-TCP, which releases calcium to promote bone growth while being more degradable and osteoconductive. Compared to PCL alone, PCL/β-TCP has a better potential to replace bone, stimulate bone regeneration, and assist bone formation [4]. Composite scaffolds of PLA and n-HA were used with different percentages to induce osteogenesis in a rabbit model. The result indicated that the PLA/15% n-HA composite scaffold could maintain biological activity as well as suitable mechanical properties in the defect of the rabbit model [40]. Furthermore, designing a prosthetic device solely out of polymeric materials may appear to be feasible (because of their low elastic modulus), but their weak strength renders them unsuitable. Metallic prosthetic limbs, on the other hand, typically fail to meet surface compatibility standards. Thus, most modern limb prostheses are made of polymer-based composites. These materials are desirable due to their high strength-to-weight ratios and high biocompatibility [34].

## 4. 3D Printing Techniques

There is a broad classification of the 3D printing techniques and their operating principles. In biomedicine, 3D printing technology can be categorized according to the manufacturing technique. The types of AM technology mainly used in the medical field include SLA, FDM, and powder bed fusion (PBF) [11,16].

### 4.1. Stereolithography Apparatus

The first 3D printing technology introduced in medicine was SLA. In SLA’s liquid-based 3D printing technology, an ultraviolet laser beam is used to selectively harden the photo resin in layers. Each layer is solidified, and successive layers are created on the top until the 3D object is formed. SLA has become a valuable tool for developing biocompatible scaffolds due to its ability to integrate bioactive materials and create specific internal structures and external geometries. SLA has been developed for ceramic-based printing by applying ceramic powders to photosensitive resin. The whole cross-sectional area of the model is scanned, and the produced layer is solid. When a layer is finished, the build platform moves down, and a sweeper blade coats the surface with an additional layer of ceramic suspension [11,14,16].

### 4.2. Binder Jetting

Binder jetting (BJ), also known as inkjet 3D printing, is compatible with powdered metals, polymers, and ceramics. Inkjet technology is used in the process to deposit a liquid binder on the powder particles [11]. Polymer gluing or a hydraulic setting reaction can be used to solidify the powder particles. The latter is known as low-temperature 3D printing, and it involves injecting a reactive liquid solution into a CaP powder base [10]. No additional thermal treatment is required during the printing process, allowing for local deposition of polymers and biologically active drugs. By combining MSCs with osteoconductive scaffolds, this technique allows for the development of cell-based bone grafts that can improve bone regeneration. Inkjet technology has shown good cytocompatibility and has been found to be appropriate for cell printing. Living cells are floated in ‘bioinks’, made up of water, polymers, or hydrogels, and then printed utilizing a thermal or piezoelectric inkjet technique [41].

### 4.3. Extrusion-Based Printing

Fused deposition modeling (FDM) and direct ink writing (DIW) are two extrusion-based methods [11]. Extrusion-based printing has some benefits such as high cell density (>1 × 10^6^ cells mL^−1^ or even spheroids) and high resolution (100 µm). However, the main disadvantage of extrusion-based printing is shear stress during the fabrication process leading to cell death [42].

#### 4.3.1. Fused Deposition Modeling

FDM is a method of additive manufacturing where layers of materials are fused together in a pattern to create an object [11]. In solid-based 3D printing, tiny beads of melted thermoplastic materials are extruded from a small nozzle and hardened afterward to form layers and then 3D parts. In this process, the material is melted at approximately 200 °C in a heating chamber and then extruded through a nozzle on the build platform one layer at a time. The printing chamber and platform are kept at temperatures below the melting point of the material but higher than room temperature to promote adhesion to the printed bed and to reduce thermal stress [11,14]. The commonly 3D-printed materials by FDM technology utilized for BTE are thermoplastic polymers (e.g., PLA, PLGA, PDL, ABS) and composites (e.g., PLGA/HAp nanofibers) [11,14,43]. The use of an FDM printer has allowed the orthopedic surgeon to print 3D models for clinical cases with very low cost, and depending on the size of the model, the cost of a model rarely exceeds USD 5 [16].

#### 4.3.2. Direct Ink Writing

Direct Ink Writing (DIW) is the most common technique for 3D printing bioceramic scaffolds. It is beneficial to produce porous structures such as scaffolds for tissue engineering and other biomedical applications. DIW employs a dispensing system to extrude a liquid-phase ink containing a high-volume content of ceramic powder through a nozzle or syringe, layer by layer, following a digitally specified pattern to produce a 3D construct [11]. Although this technology was initially designed to print polymers, it could also be used to create bioceramic or metal scaffolds. Studies showed DIW to effectively develop pure printed porous Ti6Al4V, HA, and silica-carbon-calcite composite scaffolds for biomedical applications [7,44,45,46]. The main advantages of DIW are that it can be used with a wide range of bioceramics and that the pore size, pore orientation, and lattice architecture of the printed scaffold can all be controlled. Furthermore, it is a fast, versatile, and cost-effective approach that the material paste could be deposited by varying the syringe pressure, thus allowing printing with low or without heat [7]. Previous studies showed silica-carbon-calcite scaffolds to be successfully 3D-printed by DIW (Figure 2a–c). The proposed scaffolds could serve as promising candidates for BTE applications due to their simplicity of the processing and the outstanding mechanical performances [44,46].

### 4.4. Powder Bed Fusion

PBF is an AM process in which thermal energy selectively fuses regions of a powder bed. There are different PBF methods that use a powder bed and a focused laser (selective laser melting (SLM), or sintering (SLS)) or an electron beam (electron beam melting (EBM)) to fuse powder particles by thermal energy [11]. During PBF, the focused energy scans each layer according to the cross-section generated from the STL file of the fabricated 3D object, and a one-layer thickness of loose powder is smoothly spread over a build platform. The powder is melted or sintered to merge with the surrounding material. Then, the build platform is lowered, and layers of material are applied on top until the fabrication is completed. The sintering process intrinsically leads to a porous internal structure and a rough surface since the powders are not melted completely, while the melting process consolidates the powders and thus creates parts with a higher density and improved mechanical properties [11,14]. Many materials can be processed by PBF, including plastics, metals, and ceramics, but only a single material can be utilized in the final part. Additionally, powder-based technologies are the most favorable methods as they offer high product quality and a wide range of biocompatible and implantable materials, such as nylon, PEEK, UHMWPE, SS (316, and 316 L), Ti6Al4V alloy, CoCr alloy, and ceramics [11]. Unlike SLA and FDM technologies, the model being constructed by SLS, or EBM does not require supporters because it is always surrounded by unsintered powder. Depending on the type of power source, PBF can be further divided into two major printing techniques: (1) EBM, and (2) SLM or SLS [14,35].

#### 4.4.1. Electron Beam Melting

In EBM, a high-power electron beam is used to melt and fuse the powdered metal particles. The electron beam scanning process is performed in a vacuum chamber; thus, EBM can be harnessed to process materials that are highly reactive in oxygen with reduced cost and superior speed. Metal alloys and ceramics have been printed using EBM [11,12]. Several metallic powders are currently available for EBM, including Ti6Al4V, and Co-Cr alloys. Biomedical devices such as customized orthopedic implants and instruments with complex structures and parts, including porous geometry and specified stiffness, can be manufactured by EBM. This process yields objects with slightly rough surfaces that can be advantageous for implants because they promote bone adhesion [11,14].

#### 4.4.2. Selective Laser Sintering

A high-power laser beam fuses the powdered materials in a layered fashion to form a 3D object. Laser pulses heated a powder material to achieve partial melting in the SLS technique [11,14]. Controlling the temperature in each layer to optimize the relative porosity and quality allows for producing objects with complex geometries and high dimensional accuracy. The powder properties influence the selection of an appropriate SLS laser (particle size, powder composition, and mixing). SLS can fabricate complex structures, such as BTE scaffolds, from a diverse range of powder materials, including polymers (e.g., PLA, PCL, polyamide, nylon, polystyrene, and polypropylene), ceramics (e.g., HA), and metals (e.g., titanium alloy, Cr-Co alloy, and SS) [11,16,31]. A special subtype of this technology is selective laser melting (SLM) that uses a high-energy laser to fully melt the materials. Materials used in this process are primarily metal alloys and ceramics [11].

## 5. Application of 3D Printing in Veterinary Orthopedics

The 3D printing techniques are used in the biomedical industry for the manufacturing of (1) personalized anatomical models, (2) receiver-specific surgical instruments, (3) custom-made surgical implants and prosthetics, and (4) tissue engineered scaffolds [11,14]. Despite the advancements of 3D printing in personalized medicine in humans, its applications in veterinary traumatology and orthopedics are limited and mainly consist of case studies [8]. Nowadays, replacement surgeries, corrective osteotomies, arthrodesis, critical size bone defects, limb sparing surgeries, and complex fractures are among the procedures in companion animals, which already necessitate the implant design, manufacturing, and individualization [8,12]. The AM can offer several advantages that might affect veterinary orthopedics in future [11,14,16]. Figure 4 provides a summary of the application of 3D printing in veterinary orthopedics.

### 5.1. 3D-Printed Anatomical Models

In human medicine, 3D-printed models are currently the standard of care for many complicated orthopedic problems and uncommon instances [8,14]. Although the use of 3D-printed models in veterinary medicine is less frequent than in human care, similar technologies have been used to manufacture anatomical models for: (1) pre-operative planning and surgical rehearsal, (2) veterinary research and education, and (3) improving client communication [8,13,16].

### 5.2. Pre-Operative Planning and Rehearsal

The invention of computer representations of a receiver’s anatomy that are adapted to receiver-specific data is known as receiver-specific simulation or virtual model [14]. Receiver-specific simulation is gaining popularity due to its ability to improve personalized treatment by analyzing surgical risks and outcomes. Studying the 3D virtual or physical models will enable the surgeon to understand the receiver-specific anatomy and related pathology, facilitating the selection of receiver-specific treatment choices [11,14]. Moreover, the models enable the surgeon to rehearse the surgical procedure on a replica before performing it on the actual receiver. In this manner, the surgeon can enter the surgical theater with more practice and confidence, which in turn could improve the receiver safety [8,11,16]. There is a variety of materials available to use for fabricating a 3D model. ABS models are tough, but they can be difficult to saw or drill, especially in models of larger dogs. Thus, plaster-based and ABS models can be used for diagnostic purposes. SLA models are long-lasting and simple to saw or drill and are usually utilized in diagnostic and surgical practice [8]. Three-dimensional models constructed from PLA can even be sterilized for use in the surgical setting in the case that the surgeon needs to refer to the receiver-specific anatomy and manipulate the model for performing surgical procedure [14,16]. By facilitating preoperative planning and plate contouring, as well as intraoperative referring, 3D models can also help a variety of surgical procedures, including minimally invasive plate osteosynthesis (MIPO) and open-but-do-not-touch (OBDNT) techniques for fracture management. In this manner, the pre-contoured plate can assist in fracture reduction and the procedure can be performed faster with minimal intraoperative manipulation [14,45,47,48,49]. Three-dimensional computer-assisted surgical planning and modeling were used in repairing a complex distal femoral fracture with articular engagement in a dog [50].

### 5.3. Veterinary Research and Education

Models can also be used for the purpose of education and training. Models are useful for discussing surgical approaches as well as consulting the treatment procedure between surgeons. As performing many complex orthopedic surgeries has a long learning curve, the junior surgeon can rehearse the surgical procedure on a replica several times to boost their surgical skills before performing it in the real clinical setting. In addition, 3D models can serve as educational tools in universities’ teaching hospitals and advanced orthopedic training courses for training the veterinary students and young surgeons. Recently performed veterinary research showed the importance of 3D modeling for enhancing undergraduate students’ learning experiences. The results showed that the students who used 3D models had a greater understanding of anatomical concepts and their interactions than students who used computer models or textbooks [11,14]. Veterinary research has also benefited from RP technology [16].

### 5.4. Client Communication and Education

Physical models have been shown to increase the receiver satisfaction and to aid communication with receivers in human medicine by providing a better visual understanding of their actual medical condition [11,14]. This is also so crucial in veterinary practice since the medical condition and the surgical procedure should be explained to the pet owner [8,16]. It is anticipated that prototypes will lead to improvement in client communication and education in veterinary medicine by allowing owners to better comprehend the surgical complexity, possible complications, and associated cost of the procedures. For instance, in corrective osteotomy surgeries, biomodels were found to be very effective for owner education by providing a visual depiction of both the angular limb deformity and the intended correction [16,49].

### 5.5. Receiver-Specific Orthopedic Instruments

Three-dimensional modeling can be used for designing and manufacturing special tools for orthopedic surgery on a receiver-specific basis [14]. Customized orthopedic instruments are used for improving the precision and accuracy of a particular surgical procedure. This may include facilitating the placement of gigs and implants as well as guiding an osteotomy saw or drill in an exact direction. The use of receiver-specific instruments (PSIs) has been reported extensively in human medicine for osteotomy accuracy in bone tumor resection surgeries, for accurate implant placement in total joint replacement (TJR) surgeries, and for corrective osteotomies in the treatment of limb deformities [11,14]. In veterinary medicine, a variety of materials has been used to make PSIs (Table 1). Cutting and drilling guides are useful to perform corrective osteotomies for angular limb deformities, limb sparing surgeries for tumor resection, TJRs, and arthrodesis surgeries in animals [8,11,12,13,14,47,48,51]. In one study, for example, a cemented canine total knee replacement (TKR) system was implanted in 24 dogs, using PSIs and cutting blocks for guiding the tibial and femoral osteotomies [52]. PSIs have the benefits of improving surgical precision and reducing intraoperative settings and times, but they can have two crucial disadvantages. First, actual bone anatomy of the receiver is required for developing PSIs, which may be obtained by exact segmentation of the receiver’s imaging data. Second, precise and stable placement and proper fit of the PSIs on the desired bone structure are required to ensure the accuracy of the pre-planned surgical operation [13,14,47]. The errors in instrument placement on the bone surface might be attributed to the instrument design process in CAD software or owing to the inaccurate surgeon’s assessment, which could render instrument-guided procedures less reliable. Thus, for optimizing the usage of this valuable tool, studies are needed to make the intraoperative procedure objective and even operative-independent [13,14].

### 5.6. Custom-Made Orthopedic Implants

Three-dimensional printing technology has been increasingly employed for the creation of personalized implants. There is a vast variety of scale and geometrical variations in veterinary orthopedics; for instance, hip stems for THR surgery in humans are typically available in 6 or 7 sizes while canine hip stems are available in 11 or 12 sizes and accommodate small- to large-breed dogs [8,12]. Commercially available bone implants with standard sizes are fabricated for meeting the surgical needs for most of the veterinary receivers. However, custom-made implants are required in some cases, such as when the receiver’s anatomy deviates from normal ranges or when a better surgical outcome can be predicted with a precise fit of the implant to the bone [8,12,14]. There is a variety of materials used to make metal custom-made implants (CoCr, titanium, and SS alloys) [11]. The use of 3D-printed custom-made implants has been documented in various small animal orthopedic procedures [8,16].

#### 5.6.1. Total Joint Replacement

THR, TKR, total elbow replacement (TER), and recently, patellar groove replacement (PGR) are the TJR procedures performed in veterinary [21,35].

#### 5.6.2. Total Hip Replacement

THR is a salvage procedure to eliminate the source of pain and restore the function in a receiver suffering from osteoarthritis (OA) of the hip joint. In human medicine, indications for using custom acetabular prosthesis (CAP) contains revision surgeries, large bone loss with possible pelvic discontinuity, and insufficient bone stock. CAP has been reported to be successful for the reconstruction of catastrophic acetabular bone defects in human receiver [14,35]. The use of CAP was reported for performing THR in a dog with an acetabular bone defect after femoral head and neck ostectomy surgery. A biflanged CAP was designed and 3D-printed in Ti6Al4V ELI powder (ASTM F 3001) with a direct metal laser sintering (DMLS) printer (Layerwise; 3D Systems, Rock Hill, SC, USA) to restore the acetabular bone loss. The CAP was designed with a porous surface for long-term biologic fixation. An UHMWPE cup (BioMedtrix) was cemented into the CAP using polymethylmethacrylate (PMMA, BioMedtrix), and a bolted cementless femoral stem (BioMedtrix) was inserted [2]. Recently, a custom-made proximal femoral and hemipelvic endoprosthesis was designed for limb salvage in a dog with a malignant neoplasia (histiocytic sarcoma) of the coxofemoral joint. A 3D model was generated in CAD software based on CT images of the receiver and was used to fabricate the hemipelvic implant of titanium alloy by DMSL technique. The surfaces of all implants were coated with HA to stimulate osseointegration. The hemipelvic implant was designed for anchorage to the ilium and ischium with screws, and its abaxial surface provided a textured acetabulum for cementing an UHMWPE cup (Biomedtrix). The proximal femoral endoprosthesis for cementing into the medullary canal was fabricated with porous cupola and was coated with HA to allow for tendon attachment. The results of the study supported satisfactory levels of limb function and improving the receiver’s quality of life [1]. Total hip stems and cups for dogs have been manufactured commercially using metal additive technology (BFX implants; BioMedtrix, Boonton, NJ, USA). However, joint prostheses are better manufactured with the aid of modeling. Low-modulus canine femoral stems made by EBM were engineered, and their stability was tested in vitro [8].

#### 5.6.3. Total Knee Replacement

Surgical intervention for severe OA of the stifle joint is mainly limited to stifle joint arthrodesis or limb amputation, neither of which preserve the animal’s quality of life. Although TKR is a very common and successful surgery in human receivers with knee OA, its application in veterinary is limited, and no long-term study is available yet. Despite the development of commercial canine TKR implant (Canine Total Knee, BioMedrix) and its application in more than 350 cases worldwide, the implant system did not accommodate TKR in small-breed dogs. Therefore, custom-made TKR remain an invaluable component of the TKR procedure [35,53]. The first custom TKR procedure was performed in 2005 in a dog to manage a medial femoral condylar nonunion secondary to a gunshot injury. Based on the receiver’s CT scan images, an SLA model of the distal portion of the femur was developed. A 3D replica was employed to design and manufacture a custom augment for substitution of the medial femoral condyle defect and to construct a custom femoral stem for cemented intramedullary condylar fixation. The two custom-made parts were adapted to the geometries of the previously available canine TKR system, comprising a CoCr femoral component and a monobloc UHMWPE tibial bearing surface component (BioMedtrix). The condylar augment was fabricated in Ti6Al4V and porous tantalum and was snuggled in the femoral component to restore the missing medial femoral condyle. A femoral medullary stem of Ti6Al4V was fabricated and added to the femoral component. The short-term follow-ups showed successful outcomes regarding management of a severely abnormal stifle joint using cemented canine TKR [53]. Over the past few years, there has been increased interest in the advancement of revision options for canine TKR, as well as personalized implants that may be used to treat animals with severe knee joint dysfunction or dogs with malignant lesions in the distal femur or proximal tibial. Despite the lack of case sequence, isolated case reports show that hinged implants can be used in these difficult situations. However, evidence on the long-term survival of these implants is lacking [35]. Just one study of TKR in a cat has been recorded. In this retrospective case series study, the clinical application and outcome of custom-made TKR were studied in nine cats. The implant in this case was a custom-made constrained uniaxial and rotating hinge TKR. However, additional cases and research are required to evaluate the feasibility of TKR in cats [54].

#### 5.6.4. Patellar Groove Replacement

PGR is a novel surgical procedure that includes replacing the femoral trochlear groove with a PGR prosthesis. This technique has recently been reported as a salvage procedure in conditions of severe OA of the trochlear groove due to patella luxation. There are few studies on the use of PGR, and no reports on custom-made PGR are available in the literature [55,56]. In a retrospective case series study, thirty-five dogs with patellar luxation in association with extreme patellofemoral OA had their femoral trochlear groove replaced with a PGR prosthesis (KYON AG, Zurich, Switzerland). The PGR prosthesis was made up a foundation plate and a trochlear implant. The foundation plate is made of perforated grade 4 titanium that has been coated with an anodized glow discharge and incorporates CaP. The trochlear prosthesis was manufactured with a grade 5 Ti6Al4V and coated with amorphous diamond-like carbon to minimize friction and have a scratch-resistant surface. The study’s findings revealed that using PGR could reduce OA-related lameness and increase patella stability and proper extensor mechanism alignment [56].

#### 5.6.5. Total Elbow Replacement

Since the 1970s, TER has become a successful therapy for OA diseases of the elbow in humans. Despite the advancements in veterinary joint replacement, TER is not still the treatment of choice in dogs with elbow conditions [51]. A short-term, prospective clinical evaluation of a TER system was performed in twenty large-breed dogs with naturally occurring elbow OA. The results showed that TER can be considered as a treatment alternative for adult dogs with lameness due to severe elbow OA. The drilling and bone-cutting guides were designed and used for preparing the humerus for the humeral component implantation. These drilling and bone-cutting guides showed to be valuable for implant positioning and to facilitate the accuracy of implantation. Results from this study have led to further modifications in the component design and surgical procedure of the canine TER system [3,57]. Constrained (hinge-like) TER implants were utilized with moderate effectiveness in veterinary medicine at first. Later, a non-constrained TER system was created for use in dogs with chronic elbow OA (BioMedtrix; Boonton, NJ, USA). Some modifications were adapted in designing the radioulnar and humeral components and in surgical techniques. The humeral stem was redesigned for composite fixation. Additional modifications were made in the contours of the articulating surfaces and were integrated in the humeral component to increase the range of motion [45]. Several authors have recorded improvements in the implant design and surgical techniques to the TER system in veterinary medicine [3,30,45]. Nowadays, the TER systems mainly include semi-constrained TER implant designed by Conzemius and marketed as the Iowa State elbow (BioMedtrix); cementless semi-constrained TER system (TATE Elbow; BioMedtrix) developed by Acker; new semi-constrained implant Sirius elbow (Osteogen Ltd., Bristol, UK), developed at the University of Liverpool [3,57]. In addition, the canine unicompartmental elbow (CUE) Arthroplasty System^®^ (Arthrex Vet Systems, Naples, FL, USA) has been created to address medial compartment disease of the elbow in dogs and has been tested in 103 canine cases (prospective multicenter case series study) with favorable outcomes [35,58]. Despite the developments in TER, in situations of severe bone loss, it might be challenging to restore the elbow joint’s anatomy and function with conventional TER. The construction of a 3D-printed custom-made elbow prosthesis may be a viable option in these cases [35,57,59]. Until now, no single report or study on the use of custom-made TER implants in dogs and cats has been published. A 3D-printed custom-built TER constructed of titanium alloy was recently employed in a human receiver with severe distal humerus bone loss, and the results were encouraging [59].

#### 5.6.6. Limb-Sparing Surgery

Although limb amputation is still the most common surgical choice for dogs with appendicular bone tumors such as OSA, limb-sparing surgery might help the receivers by conserving their quality of life. Limb-sparing surgery is a salvage technique which involves removing the primary tumor of bone and applying internal or external fixation to the remaining bones, with or without segmental bone substitution. Historically, the most frequent distal radial limb sparing approach was carried out using an allograft to fill the bone defect generated by segmental osseous excision [4,12,60]. In limb-sparing procedures, 3D printing technologies have been used to restore bone defects and replace substantial areas of bone loss [4,12]. The advantages of 3D printing in limb-sparing procedures include precise 3D planning of tumor resection for oncological clearance as well as designing receiver-specific cutting tools and custom-made implants for perfect reconstruction of the bony defect. These preparations will aid in shortening the intervention’s duration and lowering the associated expenses. The receivers’ post-operative prognosis, recovery, and implant-related failures will also be expected to improve [12]. Three-dimensional printing is a valuable technique in human medicine for guiding bone tumor resection and designing custom-made implants for bone reconstruction in tumor surgeries [14,15,61]. In veterinary, a personalized AM titanium implant was used in four large-breed dogs with appendicular OSA. Based on CT data, 3D models of the damaged bones were constructed. These digital models were used to make personalized titanium implants and special surgical support tools required to restore the bone defect. Manufacturing of 3D-printed cutting and drilling guides was performed using the SLA technique, from a Nextdent Dental SG material. The 3D-printed receiver-specific titanium alloy implants were designed and manufactured using two PBF techniques, EBM and SLM, from Ti6Al4V powder, and were successfully applied in canine cases following limb-sparing surgery [12]. The largest report of clinical series in veterinary has been performed in six dogs with mandibular tumors, five dogs with distal radius tumors, and one dog with distal tibia tumor investigating the clinical outcomes of using receiver-specific 3D-printed titanium endoprostheses. Each osteotomy was performed with a guide of thermoplastic cutting jig designed to guarantee the accuracy of osteotomy. Custom-made implants were created using several CAD software and produced using SLM technique in titanium-6 aluminum-4 vanadium alloy using CT images of receivers [62]. Another study has evaluated the management of distal radius OSA of a mixed-breed dog by limb-sparing surgery using a custom tantalum distal radial endoprosthesis (Biomedtrix, Boonton, NJ, USA) with trabecular structure [63]. In a flat-coated retriever with a malignant neoplasia (histiocytic sarcoma) of the left femoral head and acetabulum, custom-designed hemipelvic and proximal femoral endoprostheses were employed. The implant was created using HA-coated surfaces to facilitate tissue integration and was constructed using the dog’s CT images. Screws were used to secure the pelvic implant to the ilium and ischium. The proximal femoral implant allowed for tendon ongrowth and muscle attachment, and the femoral implant was cemented [1]. In one dog with tibial OSA, a customized implant was utilized in combination with TKR to replace the proximal section of the tibia. The EBM-fabricated customized titanium implant contained porosity features for bone ingrowth and anchoring features for reattaching ligaments and tendons to restore knee joint function [8]. According to the findings of one study, personalized endoprostheses and cutting guides can reduce limb sparing surgery time by 25–50% and may lower the risk of implant failure. According to the numerical model, increasing the modulus of elasticity of an implant material from 25 to 50 GPa would improve stress distribution within the implant [64].

#### 5.6.7. Corrective Osteotomies

Pre-operative planning and surgical correction of complex angular limb abnormalities based on radiographic assessments can be difficult, especially for torsional and multi-planar deformities. Thus, CT will be the best tool for diagnosing, measuring the magnitude of the deformity and planning the corrective osteotomy. Angular limb deformities of forelimbs and hindlimbs, premature closure of growth plates, and patellar luxation due to femoral or tibial torsions are amongst conditions that can now be addressed by aiding of CT imaging, digital simulation, and AM technology [45,48,49]. The use of 3D printing for accurate surgical planning of corrective osteotomy is still at the beginning of its development in small animal practice [8,45,48]. Some cases of forelimb deformity in dogs have been reported to have been treated using RP technology [13,47,48,65,66]. In these studies, CAD software images were created from the receiver’s CT data and were used for virtual preoperative planning of the corrective osteotomy, limb manipulation of deformity correction and assembling 3D-printed bone models [13,47,48,65]. In a study, custom-made 3D-printed saw guides were fabricated to enhance the accuracy of the corrective osteotomies of antebrachial deformities in six dogs. The surgical corrective procedure contains radial closing wedge ostectomy and ulnar osteotomy. An FDM 3D-printer was used to fabricate the CAD antebrachial limb model and saw guides in ABS plastic which were then cold-sterilized for intraoperative use as the material was not autoclavable. The simulated correction of the 3D computer images was utilized for intraoperative references without referring to a physical model. The authors of this study declared that regardless of their promising results, accurate correction of rotational deformity was difficult, and they advised further development is required [47]. Similar receiver-specific osteotomy guides were designed and manufactured in PLA for surgical correction of complex antebrachial deformities (biplanar deformity with valgus, procurvatum, and external torsion of the right radius) in two skeletally mature dogs. Computer-assisted surgery was used by the researchers to improve pre-surgical planning, perform computer simulation correction, and practice surgery on 3D-printed bone models. They claimed that using receiver-specific surgical guidance enabled them to achieve a good repair of the antebrachial deformity while also reducing the surgical time [66]. In another study, in addition to receiver-specific 3D-printed osteotomy and repositioning guides, custom-made titanium plates were produced for application in antebrachial limb deformity correction in four chondrodystrophic dogs. STL models of the canine antebrachium were fabricated for surgical rehearsal and for pre-contouring of the osteotomy and reduction guide, and the 3D-printed titanium plate. In this small number of case series, the custom-printed repositioning guides and titanium plates enabled accurate corrective osteotomy of antebrachial deformities with good clinical outcomes [48]. Surgical planning in a 10 kg dog with carpal valgus and radial procurvatum in the right forelimb was performed based on radiographs using the centers of rotation of angulations (CORAs) method. Model of the limb was 3D-printed in ABS using a FDM printer and was used to aid detection of the location and magnitude of the deformity and to perform surgical rehearsal. A partial ulna osteotomy and a double cuneiform osteotomy of the radius was performed at the level of CORAs, followed by bone alignment and osteosynthesis with plates and screws [65]. A 28.4 kg Golden Retriever dog was presented with radial shortening of the left forelimb due to premature closure of the epiphyseal growth plates. Using 3D slicer, virtual corrective osteotomy simulations were performed. Cutting guide renders were placed on the affected area using 3D rendering on the virtual 3D bone models. Receiver-specific models and PSIs were fabricated in-house from the PLA filaments using FDM and a desktop FFF 3D printer, respectively. The authors demonstrated an accurate, reproducible osteotomy technique using PSI 3D technology. The results showed that PSI technology may improve osteotomy accuracy during corrective osteotomy by providing clinically acceptable margins and offers a smaller standard deviation than the freehand method [13]. In general, the CORAs method combined with CT and 3D modelling was found to be useful for planning and simulation of the corrective surgery in a case of forelimb deformities and improved the surgical efficiency in comparison to the conventional pre-operative study [65].

Successful methods for the correction of hindlimb deformities have been reported with good functional outcomes using the 3D printed models. SLA models were manufactured in epoxy resin and served as templates to aid surgical planning in four dogs (five limbs) with complex distal femoral deformities. Models were used for direct measurements of the joint orientation angles and torsion of the femur and for direct comparison to the contralateral normal limb [49]. In another study, RP was used to create 3D printing bone models to facilitate the procedure of corrective osteotomy in a small-breed dog with grade 4 medial patellar luxation with severe bilateral bone deformities of femur and tibia. The location and magnitude of CORA and the location and angle of osteotomy were measured for each bone prior to surgery using 3D modelling [45]. In both studies, surgery rehearsal and pre-contouring of the plate were performed on replicas before execution of corrective osteotomy. The pre-contoured plates were sterilized prior to their application in actual surgery [45,49]. The accuracy of corrective osteotomy surgery appears to be effectively increased by utilizing 3D-printing replica for accurate planning of the procedure [8,37]. This may also contribute to successful surgical outcomes [35,54]. Overall, based on the surgeons’ professional knowledge, RP was deemed useful to the receiver by reducing the surgical time, morbidity, and mortality, as well as lowering overall costs [16].

#### 5.6.8. Arthrodesis

Custom-made plates and PSIs, according to the author, improved the success of arthrodesis procedures by enhancing osteotomy planning and improving the better fit of the plate to the bone at a correct limb angle. In one study, pantarsal arthrodesis was performed with a customized medial or lateral SS (316 LVM) bone plate in 13 dogs [67]. A custom-made elbow arthrodesis plate was pre-contoured to 130° and applied medially in six dogs with severe OA of the elbow or for revision of previously performed TER [68].

### 5.7. Customized Scaffolds

Recent developments in computational design have made it possible to manufacture 3D scaffolds with controlled properties that can mimic natural bone characteristics of an individual [6,41]. Three-dimensional-printed implants can also be designed with scaffold structure which can enhance osteointegration. In veterinary clinical models, scaffold structures have been used as lower extremity implants as well as in limb-sparing surgeries, tibial tuberosity advancement (TTA), and the reconstruction of critical-sized bone defects [4,8,10,14,69]. Critical-sized bone defects are difficult to treat; in cases where allograft bone is unavailable for the required dimension of the defect, 3D-printed scaffolds may be considered. The scaffolds for bone substitution should have special characteristics including sufficient mechanical properties, porosity, biocompatibility, and biodegradability [6]. Interconnected porosity of the scaffold material optimizes bone healing by supporting the vascular penetration and infiltration of new bone formation. Therefore, in theory, receivers treated with 3D-printed scaffolds might be less prone to postoperative complications, such as infection and sequestra formation. Moreover, tissue ingrowth can be enhanced by acquiring a tight interface between the scaffold and host osseous tissue. To enhance the material healing properties, 3D-printed scaffolds can potentially include cells, growth factors, and vasculature [7]. Synthetic bone substitutes, on the other hand, are an excellent alternative to biological grafts in small bone defect reconstruction but are not the best option for large bone defect reconstruction due to insufficient strength to sustain the body load and insufficient neovascularization ingrowth [7].

Various in vitro studies have shown the potential application of 3D-printed scaffolds in BTE [6,22,27,33]. CaP ceramics and bioactive glasses are introduced as promising osteoinductive and osteoconductive substitutes for large orthopedic defect remolding or regeneration [4,6,69,70]. The custom-made scaffolds can be manufactured using FDM and electrospinning 3D-printing technology [69]. The potential application of a 3D-printed PLA scaffold filled with PLGA/HA nanofibers was assessed in an in vivo study on the reconstruction of bony defects in the radius of six beagle dogs. The FDM-type 3D printers (Makerbot, NY, USA) were used to build the scaffold. After creating a 20 mm-long defect in the radius bone, the 3D scaffold was replaced and subsequently fixed with an LC-DCP plate. The results indicated the biodegradability of the scaffold and its replacement by new bone tissue [69]. Three-dimensional-printed PCL/β-TCP scaffold was used in a bone defect resulting of limb-sparing surgery in a dog with distal radial OSA. Three-dimensional-printed scaffold was produced by using a mixture of powdered PCL and β-TCP and designed to have an entirely interconnecting structure. The scaffold was then manufactured through a microextrusion-based 3D printer (3DX Printer, T & R Biofab Co., Siheung, Korea), employing a heating process. The outcomes showed implant stability and increased bone opacity in host bone and scaffold interference without any complication. The authors suggested the 3D-printed PCL/β-TCP scaffold as an effective substitute to cortical allograft for the reconstruction of bony defects [4]. Another study used a 3D-printed bioceramic personalized cage implant for a TTA procedure in a large-breed dog with CrCL disease. Based on the receiver’s CT, the cage was designed with high permeability and structural integrity through a topology optimization approach. A low-temperature 3D-printing technique was employed for manufacturing the cage composed of brushite, monetite, and TCP with high porosity. The cage’s mechanical characteristics were found to be comparable to those of trabecular bone; however, there were limitations in the cage’s capacity to absorb energy. Improvements in biocompatibility and osteoconductivity of the customized cage were observed. The study proposed a modified TTA technique using the receiver-specific cage as a valid alternative to CrCL treatment in dogs, although clinical trials are required [10]. Three-dimensional-printed β-TCP scaffold combined with recombinant human bone morphogenic protein-2 were used to manage severe atrophic nonunion of the radius in a Yorkshire terrier. The results showed excellent bone regeneration and complete functional recovery [71].

Today, one of the modifications made to the implants, especially TJR implants, includes surface modification to enhance the press-fit and surface coating to promote osseous integration. For instance, in cementless Kyon hip (Kyon AG, Zurich, Switzerland) and cementless TKR implants (GenuSys Knee; Innovet, Hamburg, Germany) the addition of a plasma coating to implants has been used to promote osseous integration of the implants [35]. In one study, pseudo-greater trochanter was designed with an anatomical shape and a dome consisting of an open porous structure, and a high degree of pore interconnectivity to allow tendon attachment using sutures through the dome. The dome itself was constructed in titanium alloy to provide a biocompatible anchor for the ligaments; the remaining proximal femoral component was constructed with CoCr alloy. The implants were manufactured by Fitzbionics Ltd. (Godalming, Surrey, UK) and the plates, stipples, and rim were coated with HA to encourage osseointegration [1].

Figure 5 shows some of the current applications of 3D printing in the veterinary orthopedic field.

## 6. Limitations and Future Directions

A broad range of animal sizes and the need to anesthetize the receiver during each imaging session are two main limitations of CT imaging in animals [8,16]. In human medicine, AM enables the fabrication of personalized orthopedic prostheses from biocompatible materials such as ceramics, polymers, or metals. The use of AM in veterinary medicine is expected to grow as volumetric acquisition in MRI and 4D ultrasound becomes more common. Despite growing interest in using AM techniques in veterinary practice, applications such as veterinary orthotics, prosthetics, and reconstructive and replacement surgeries are still in their early stages [11,12,14,15,16]. AM is a promising alternative for many veterinary surgical options, such as salvage techniques, and the future of AM technology in veterinary applications is very promising. However, it may pose additional surgical challenges, such as a smaller amount of soft tissue surrounding bones, which is critical for the recovery process. To address these challenges, many traditional veterinary surgical procedures should be reconsidered, and new aspects of AM should be investigated, as has already occurred in human practice. More progress is also needed in the material, design, and manufacturing processes of PSIs, custom-made implants and prostheses that fit the specifics of veterinary receivers. Thus, expanding the use of additive manufacturing in veterinary medicine will not only benefit veterinary receivers but will also improve the understanding of individualized medicine and lead to advancements in implant and prosthesis design and manufacturing techniques in the broadest sense [12].

## 7. Conclusions

The convergence of receiver-specific simulation and 3D-printing technologies has benefited anatomic models, PSIs, and personalized implants. In veterinary orthopedics, custom implant options have been designed and used for some complex orthopedic conditions and revision surgeries. Three-dimensional technology provides this field with a valuable tool for more accurate diagnosis and preoperative planning, designing, or selecting the appropriate implant type, and performing precise surgery. Based on previously published papers, using models, PSIs, and custom-made orthopedic implants proved to be an effective method for addressing various aspects of veterinary orthopedics. The review discussed the 3D-printing techniques and materials used in veterinary orthopedics as well as their applications in small animal orthopedics.

## Figures and Tables

**Figure 1 ijms-23-01045-f001:**
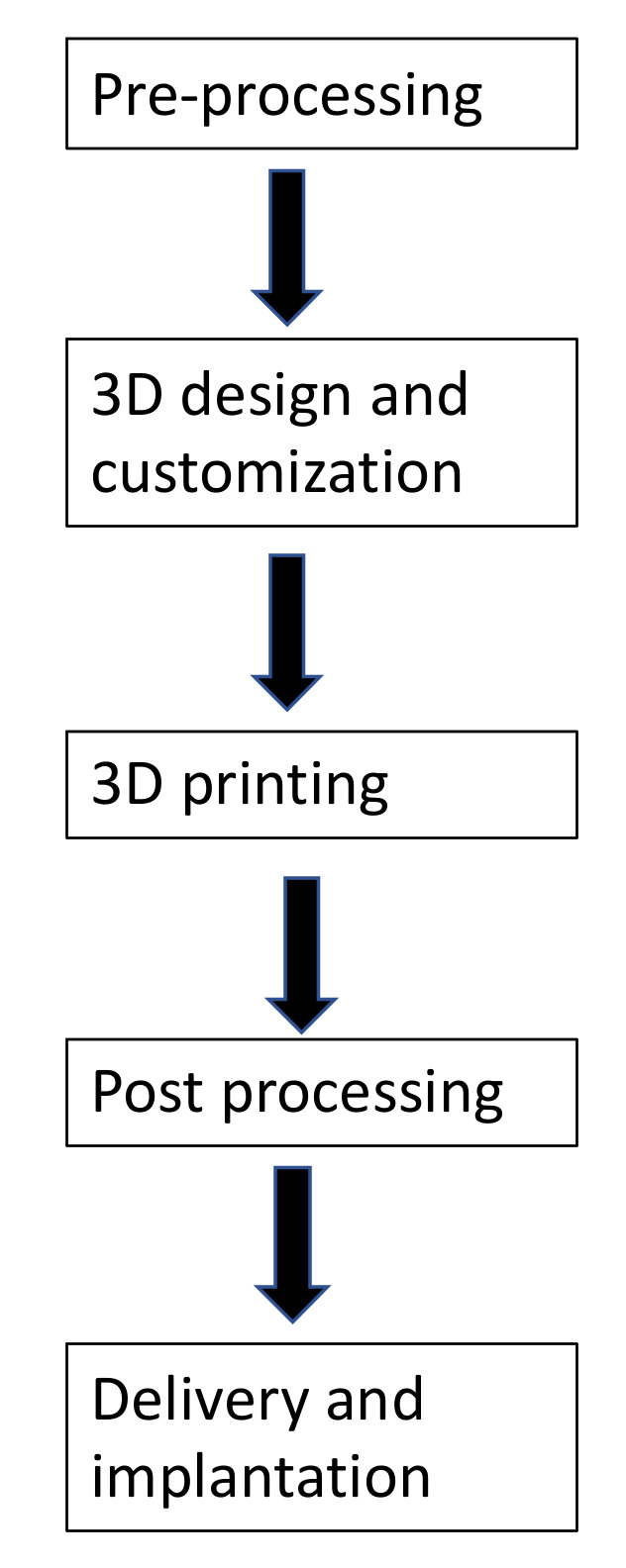
Steps for a 3D-printed project.

**Figure 2 ijms-23-01045-f002:**
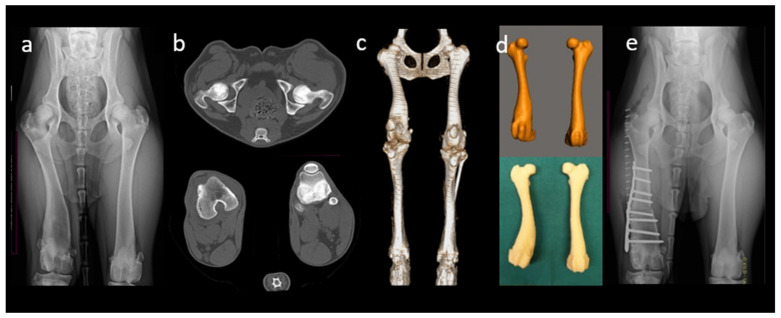
Receiver-specific approach in a 1.5-year-old Doberman with lateral patellar luxation due to complex angular limb deformity involving valgus and internal torsion of the right femur (**a**), radiographs). The steps consist of (**b**) image acquisition in DICOM from CT scans; (**c**) image processing including 3D reconstruction and volume rendering with Meshmixer software, and (**d**) 3D printing of anatomical models of both femurs using FFF technique from PLA filaments on a Delta WASP 2040 INDUSTRIAL X 3D Printer (WASP, Massa Lombarda, Ravenna, Italy). CT images and their 3D reconstruction were used to measure angular limb deformity and plan for corrective ostectomy. Models were used for studying the receiver’s anatomy, for surgical rehearsal, and for pre-contouring of the plate before actual surgery. (**e**) Using 3D modeling, medial closing wedge ostectomy and torsional correction were performed more precisely.

**Figure 3 ijms-23-01045-f003:**
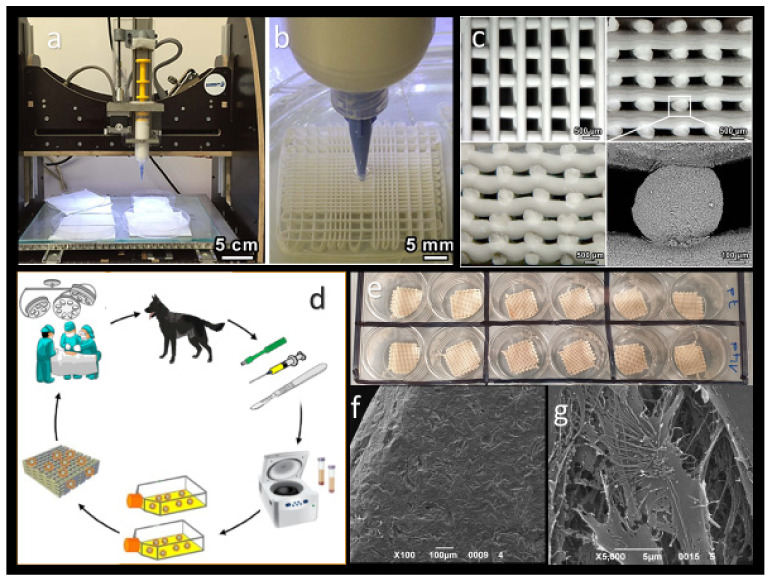
(**a**) Three-dimensional printing of silicone-based scaffold using DIW; (**b**) detail of 3D-printing process in an oil bath; (**c**) morphology of ceramic 3D-printed scaffolds from different views and high magnification detail of a rod fracture surface; (**d**) MSCs therapy steps include retrieval of adipose tissue from healthy dogs, isolation and characterization of cAD-MSCs, pelleting and seeding the cAD-MS on scaffolds, and its potential in vivo or in clinical applications; (**e**) cAD-MSCs pelleted and seeded onto the carbon-based scaffolds. SEM images of cAD-MSCs 7 days after culture on carbon-based scaffolds at magnifications of (**f**) 100× and (**g**) 5000× reveal significant secretome activity of the cell surfaces on the carbon-based scaffolds. (**a**–**c**) copyright 2022, IOP Publishing; (**e**–**g**) [6].

**Figure 4 ijms-23-01045-f004:**
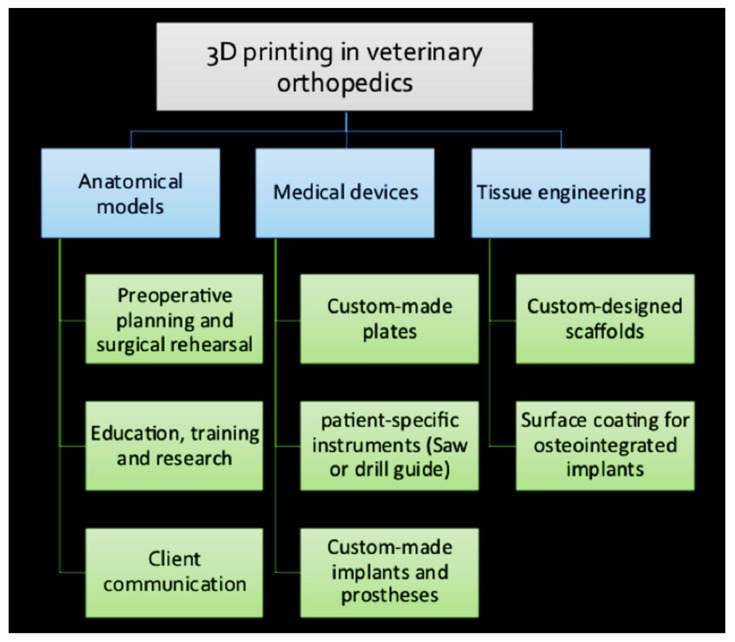
Summary of application of 3D printing in veterinary orthopedics.

**Figure 5 ijms-23-01045-f005:**
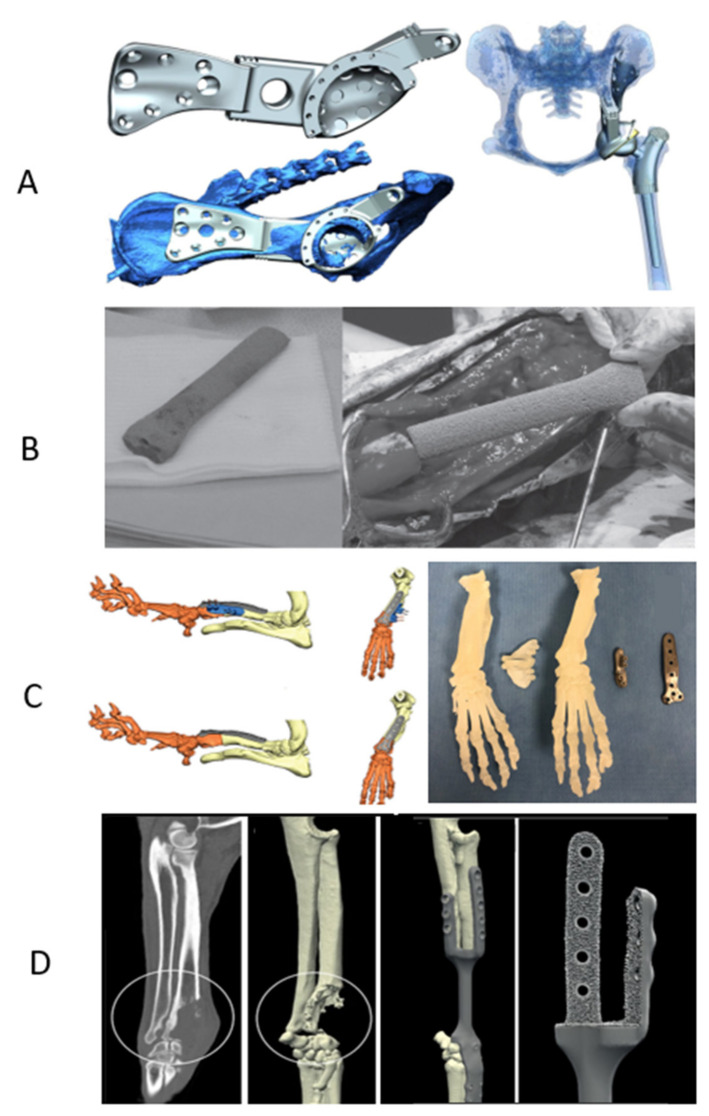
Custom-designed hemipelvic and proximal femoral endoprosthesis for limb salvage technique in a dog (**A**), custom-made tantalum distal radial endoprosthesis for limb sparing surgery in a dog (**B**), virtual planning (**C**, **left**) and custom-made 3D printed SLA models, osteotomy guide, reduction guide, and titanium plate (**C**, **right**) for correction of antebrachial limb deformity in a dog. (**D**) custom-made titanium implant for limb sparing surgery in a dog with distal radial OSA. Copy right 2022, (**A**) BLACKWELLPUBLISHING, INC., (**B**) Canadian Veterinary Medical Association, (**C**) Georg Thieme Verlag KG, (**D**) Daehanuiyongsaengchegonghakoe.

**Table 1 ijms-23-01045-t001:** Examples of 3D printable materials used for receiver-specific orthopedic application in recent publications in veterinary orthopedics.

Group	Material	3D Printed Object	3D Printing Technique	Clinical Case or Study	Surgical Intervention	Reference
Metals	CoCr with plasma coating	Femoral and tibial components of Custom-made constrained uniaxial and rotating hinge TKR	Fitzbionics Ltd. (Godalming, Surrey, UK)	9 cats with traumatic stifle luxation or severe distal femoral deformity	Custom TKR	Fitzpatrik et al., 2021
	Titanium alloy (Ti6Al4V)	Personalized limb-sparing endoprostheses	Laser PBF EOSINT M280 400 W Ytterbium fiber laser system (EOS GmbH, Munich, Germany)	In-vitro testing and modeling of a canine limb	Limb sparing surgery	Timercan et al., 2019
		Custom implant of proximal tibia (with porous features for ligaments and tendons reattachment) in conjunction with commercial TKR	EBM	Large breed dog with OSA of the proximal tibia	Limb sparing surgery	Harrysson et al., 2015
		Biflanged CAP with a porous surface for long-term biologic fixation	DMLS (Layerwise; 3D Systems, Rock Hill, SC, USA)	Adult Labrador retriever with lameness after femoral head and neck ostectomy	Custom-made THR to restore the acetabular bone loss	Castelli et al., 2019
		Custom-made limb-sparing implants	PBF including EBM and SLM techniques	Four adult large-breed dogs with OSA	Limb sparing surgery	Vladimir et al., 2019
		Custom-made plate	EBM (Arcam EBM; Designvägen 2, SE-435 33 Mölnlycke, Sweden)	Four small chondrodystrophic breed dogs with antebrachial limb deformities	Corrective osteotomy (closing wedge ostectomy of the radius)	Carwardine et al., 2020
		Custom-made hemipelvic and proximal femoral endoprosthesis (coated with HA)	DMLS	Adult flat-coated retriever dog with bone lysis of femoral head and acetabulum due to invasive histiocytic sarcoma	Limb salvage technique	Fitzpatrick et al., 2018
Ceramics/Composites	PCL/β-TCP	Custom-designed scaffold	Microextrusion-based 3D printer (3DX Printer, T&R Biofab Co., Siheung, Korea)	Adult Great Pyrenees breed dog with OSA of distal radius and ulna	Limb sparing surgery in a dog with distal radial OSA	Choi et al., 2019
	PLA/PLGA/HA	PLA scaffold filled with PLGA/HAp nanofibrous scaffold	FDM 3D printing for PLA (Makerbot, NY, USA) and electrospinning procedure for 3D electrospun nanofibrous scaffold	Bone defects (20 mm) created in radius bone of six beagle dogs bilaterally (in-vivo study)	Bone defect reconstruction surgery	Yun et al., 2019
	Brushite/Monetite/TCP	Customized TTA cage with scaffold structure	Low temperature 3D printing	Adult rottweiler dog with CrCL deficient stifle	Modified TTA	Castilho et al., 2014
	β-TCP (loaded with recombinant human bone morphogenic protein-2)	Custom-designed scaffold	DIW	Adult Yorkshire terrier dog with critical-sized bone defect of left radius	Surgical management of severe, radial atrophic nonunion	Franch et al., 2020
	HA/TCP	Customized scaffold	Digital light processing (DLP)	Twelve healthy adult beagle dogs (in-vivo study); 48 defects were created (two defects on each side of the mandible)	Scaffold placement in defect for bone regeneration	Kim et al., 2020
Polymers	ABS	Custom-made saw guide	FDM (Dimension Elite; Dimension, Inc., Eden Prairie, MN, USA)	four small- and two large-breed dogs (seven limbs) with antebrachial angular limb deformities	Corrective osteotomy (radial closing wedge ostectomy and ulnar osteotomy)	Worth et al., 2018
		Personalized cutting guides	FDM	In-vitro testing and modeling of a canine limb	Limb sparing surgery	Timercan et al., 2019
		3D model	FDM	Eight-month-old Azawakh dog with angular limb deformity of right forelimb	Corrective osteotomy	Bordelo et al., 2018
	PLA	Patient-specific models	FDM (Alpha-i3, Alpha3-D, Seoul, Korea)	Adult Golden Retriever dog with angular limb deformity	Corrective osteotomy	Lee et al., 2020
		Patient-specific cutting guides	FFF (Alpha-i3, Alpha3-D, Seoul, Korea)			
		Bone models	3D printing (Drukarka 3D, 3D Gence SP., Przyszowice, Poland)	Two adult dogs with antebrachial limb deformity	Corrective osteotomy	Longo et al., 2019
	Epoxy resin	3D Model	3D printing (Form 2 printer; Formlabs, Somerville, MA, USA)	Adult Labrador retriever with lameness after femoral head and neck ostectomy	Custom-made THR to restore the acetabular bone loss	Castelli et al., 2019
		3D Model	SLA (Form 2: Formlabs, Somerville, MA, USA)	Adult Golden Retriever dog with severely comminuted fracture of distal femoral supracondylar and bicondylar region	Surgical repair of complex femoral articular fracture	Lam et al., 2019
		3D biomodels	SLA	Four dogs (five limbs) with complex distal femoral deformity	Corrective osteotomy	DeTora et al., 2016
	Polyamide 12	Custom-made osteotomy guide	3D printing (Drukarka 3D, 3D Gence SP., Przyszowice, Poland)	Two adult dogs with antebrachial limb deformity	Corrective osteotomy	Longo et al., 2019
	UHMWPE	Cylindrical bearing (bushing) placed medial and lateral in femoral component and then on tibial component	Fitzbionics Ltd. (Godalming, Surrey, UK)	Nine cats with traumatic stifle luxation or severe distal femoral deformity	Custom TKR	Fitzpatrik et al., 2021
		Acetabular cup cemented to the hemipelvic component	(Biomedtrix, Boonton, NJ, USA)	Adult flat-coated retriever dog with bone lysis of femoral head and acetabulum due to invasive histiocytic sarcoma	Limb salvage technique	Fitzpatrick et al., 2018
	Nextdent Dental SG material	Custom-made cutting and drilling guides	SLA	Four adult large breed dogs with OSA	Limb sparing surgery	Vladimir et al., 2019
Bone cement	PMMA	Implant fixation and fill bone-implant voids				Liska et al., 2007
	Calcium carbonate/polyol-based cement	fill bone-implant voids and decrease stress of bone–implant interfaces				

## Data Availability

Data are contained within the article.

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
