# Peer review of "Active Materials for 3D Printing in Small Animals: Current Modalities and Future Directions for Orthopedic Applications"

_ijms, 2022, doi:10.3390/ijms23031045_

Round 1

Reviewer 1 Report

Dear Authors,

The review prepared by you is set up in a very good way and it is very pleasant to read. I have a few minor comments about the text - please respond to them:
First of all - I cannot fully agree to the introduction of machining processes to the table containing examples of 3D printing. Machinig is a common subtractive manufacturing technology. Unlike 3D printing, the process typically begins with a solid block of material (blank) and removes material to achieve the required final shape, using a variety of sharp rotating tools or cutters. Additive Manufacturing (AM) or 3D Printing processes build parts by adding material one layer at a time.

Secondly - I think that a bit more attention should be paid to materials - two sentences about titanium and one titanium alloy in this review is definitely not enough - maybe you should make a critical characterization of several titanium alloys used as powders in additive manufacturing. And it is necessary to describe the methods of surface modification of metal biomaterials in more detail.

"Porous metallic biomaterials are frequently used as coatings..." - section 3.1 - according me porous metallic biomaterials are not used as coatings, they are just biomaterials used in differetn paths of orthopaedy. Please consider this sentence.

Editorial error - 4.3. D printing techniques  - it should be 4. 3D printing techniques.

Author Response

The review prepared by you is set up in a very good way and it is very pleasant to read. I have a few minor comments about the text - please respond to them:

First of all - I cannot fully agree to the introduction of machining processes to the table containing examples of 3D printing. Machinig is a common subtractive manufacturing technology. Unlike 3D printing, the process typically begins with a solid block of material (blank) and removes material to achieve the required final shape, using a variety of sharp rotating tools or cutters. Additive Manufacturing (AM) or 3D Printing processes build parts by adding material one layer at a time.

Thanks to the referee comments, we changed it.

Secondly - I think that a bit more attention should be paid to materials - two sentences about titanium and one titanium alloy in this review is definitely not enough - maybe you should make a critical characterization of several titanium alloys used as powders in additive manufacturing. And it is necessary to describe the methods of surface modification of metal biomaterials in more detail.

Thanks to the referee comments, we changed it.

"Porous metallic biomaterials are frequently used as coatings..." - section 3.1 - according me porous metallic biomaterials are not used as coatings, they are just biomaterials used in differetn paths of orthopaedy. Please consider this sentence.

Thanks to the referee comments, we changed it.

Editorial error - 4.3. D printing techniques  - it should be 4. 3D printing techniques.

Thanks to the referee comments, we changed it.

Reviewer 2 Report

Authors discussed the 3D printing techniques and materials used in veterinary orthopedics, as well as their applications in small animal orthopedics. The authors cite sufficient original articles published in well-known journals on the relevant subject. The described models are adequate and can be reproduced by other researchers. The review will be of interest to researchers who study optimization of 3D technology provides this field with a valuable tool for more accurate diagnosis and preoperative planning, designing, or selecting the appropriate implant type, and performing precise surgery. I would like to recommend to revise the following notes in order to improve the quality of the manuscript:

  1. There is a little confusion with abbreviations. For example, for the first time DICOM is in the captions to Figure 1, and its interpretation is only in section 2.1. Also, not all abbreviations are decoded. For example, 3DP (in table 1) or ME3DP (in line 347). This makes it difficult to understand the text.
  2. 2.1. Image segmentation – the text is very complicated and difficult to understand. The attendant figure could help to understand the text.
  3. There is an extra colon in the heading of the paragraph 2.3. Object Fabrication:
  4. The quality of Table 1 and Figure 3 is poor.
  5. There is paragraph 3.3.1 but the paragraph 3.3.2 is not following. I recommend to replace 3.3.1 with 3.4 and so on. Also there is a mistake: instead of paragraph number 4.3 sould be 4.0
  6. Is it necessary to separate paragraph 4.4.3. Selective Laser Melting (SLM) as a particular method if there is only one reference to it. It seems to be a modification of Selective Laser Sintering
  7. Regarding the entire text: the authors need to distinguish the terms of “patient” and “veterinary patient” and use them throughout the article, as in some parts of the text the term is used in relation to a human.

I recommend the article for publication in the journal «IJMS» after making the recommended changes.

Author Response

  1. There is a little confusion with abbreviations. For example, for the first time DICOM is in the captions to Figure 1, and its interpretation is only in section 2.1. Also, not all abbreviations are decoded. For example, 3DP (in table 1) or ME3DP (in line 347). This makes it difficult to understand the text.

Thanks, we changed

  1. 2.1. Image segmentation – the text is very complicated and difficult to understand. The attendant figure could help to understand the text.

Thanks, we changed

  1. There is an extra colon in the heading of the paragraph 2.3. Object Fabrication:

Thanks, we changed

  1. The quality of Table 1 and Figure 3 is poor.

Thanks, we changed

  1. There is paragraph 3.3.1 but the paragraph 3.3.2 is not following. I recommend to replace 3.3.1 with 3.4 and so on. Also there is a mistake: instead of paragraph number 4.3 sould be 4.0

Thanks, we changed

  1. Is it necessary to separate paragraph 4.4.3. Selective Laser Melting (SLM) as a particular method if there is only one reference to it. It seems to be a modification of Selective Laser Sintering

Thanks, we changed

  1. Regarding the entire text: the authors need to distinguish the terms of “patient” and “veterinary patient” and use them throughout the article, as in some parts of the text the term is used in relation to a human.

Thanks, we changed